



# Overflow of cold water across the Iceland-Faroe Ridge through the Western Valley

Bogi Hansen[1], Karin Margretha Húsgarð Larsen[1], Steffen Malskær Olsen[2], Detlef Quadfasel[3], Kerstin Jochumsen[3], Svein Østerhus[4]

[1]Faroe Marine Research Institute, PO Box 3051, FO-110 Tórshavn, Faroe Islands
[2]Danish Meteorological Institute, Lyngbyvej 100, 2100 Copenhagen, Denmark
[3]University of Hamburg, Bundesstrasse 53, 20146 Hamburg, Germany
[4]Uni Research Climate, Nygårdsgata 112, N-5008 Bergen, Norway

*Correspondence to*: Bogi Hansen (bogihan@hav.fo)

**Abstract.** The Iceland-Faroe Ridge (IFR) is considered to be the third-most important passage for dense overflow water from the Nordic Seas feeding into the lower limb of the Atlantic Meridional Overturning Circulation with a volume transport on the order of 1 Sv ($10^6$ m$^3$ s$^{-1}$). The Western Valley, which is the northernmost deep passage across the IFR, has been presumed to supply a strong and persistent overflow (WV-overflow), contributing a large fraction of the total overflow across the IFR. However, prolonged measurements of this transport are so far missing. In order to quantify the flow by direct measurements, three instrumental packages were deployed close to the sill of the Western Valley for 278 days (2016–2017) including an Acoustic Doppler Current Profiler at the expected location of the overflow core. The average volume transport of WV-overflow during this field experiment was found to be less than 0.03 Sv. Aided by the observations and a two-layer hydraulic model, we argue that the reason for this low value is the inflow of warm Atlantic Water to the Norwegian Sea in the upper layers suppressing the deep overflow. The link between deep and surface flows explains an observed relationship between overflow and sea level slope as measured by satellite altimetry. This relationship, combined with historical hydrographic measurements allows us to conclude that the volume transport of WV-overflow most likely has been less than 0.1 Sv on average since the beginning of regular satellite altimetry in 1993. Our new direct measurements do not allow us to present an updated estimate of the total overflow across the IFR, but they indicate that it may well be considerably less than 1 Sv.

## 1 Introduction

Overflow of cold, dense water from the Nordic Seas across the Greenland-Scotland Ridge has long been recognized as an important component of the world ocean circulation (Saunders, 2001). Together with water entrained after crossing the ridge, it forms the main component of North Atlantic Deep Water (Dickson and Brown, 1994; Hansen et al., 2004), the deep limb of the Atlantic Meridional Overturning Circulation (AMOC). By transporting carbon dioxide (Sabine et al., 2004) and



heat (Hansen et al., 2016) from the atmosphere into the deep ocean, the overflow is also an important component of the climate system.

The Iceland-Faroe Ridge (IFR) is the central part of the Greenland-Scotland Ridge and it separates the deep parts of the Norwegian Sea from the Iceland Basin and the rest of the Atlantic Ocean (Fig. 1a). The occurrence of overflow across the
5 IFR, "IFR-overflow", was recognized more than a century ago (Knudsen, 1898) and there is a long history of research on the topic as reviewed by Hansen and Østerhus (2000). These studies have demonstrated that overflow occurs in many locations along the IFR and cold, dense water is found in the bottom layer all along the crest of the ridge, as exemplified in Fig. 1b.

Overflow is, however, not the only flow to cross the ridge. Except for the region close to Iceland, most of the water column above the bottom layer over the IFR is dominated by warm water (Fig. 1b) originating farther south in the Atlantic
Ocean and flowing into the Norwegian Sea. Strictly speaking, the whole region (including the Norwegian Sea) is part of the Atlantic Ocean but we will adopt the common practice in studies of this region of reserving the term "Atlantic" for the region southwest of the ridge and use the term "Atlantic Water" for this warm water.

According to the float experiment by Rossby et al. (2009), the inflow of Atlantic Water, "Atlantic inflow", across the IFR may occur in many locations, but mainly close to the two ends of the ridge, which is consistent with a map of the Mean
Dynamic Topography for the region (Fig. 2). The barotropic footprint of the Atlantic inflow therefore has to be taken into account when considering the forcing of the IFR-overflow.

For the two main overflow branches from the Nordic Seas that pass through the Denmark Strait and the Faroe Bank Channel, respectively, observational efforts have resulted in strong constraints on their volume transports (Jochumsen et al., 2017; Hansen et al., 2016). In contrast, the long history of research on IFR-overflow has not resulted in a well constrained
estimate of its volume transport. Based on the first overflow expedition (Overflow '60), coordinated within the International Council for the Exploration of the Seas (Hermann, 1967), the IFR-overflow has been considered to be around 1 Sv (1 Sv = $10^6$ $m^3 s^{-1}$) in recent literature (e.g., Olsen et al., 2016). This transport estimate was supported by Perkins et al. (1998) who reported an overflow of at least 0.7 Sv. It is also consistent with the estimate from a comprehensive autonomous glider experiment 2006–2009, which gave a lower bound of 0.8 Sv for the total IFR-overflow (Beaird et al., 2013).

The crest of the IFR is deepest close to the Faroes (Fig. 1b), but direct current measurements from the three ADCP (Acoustic Doppler Current Profiler) moorings reported by Østerhus et al. (2008) did not show strong overflow in that region and Beaird et al. (2013) estimated only 0.3 Sv to cross over the southeastern half of the ridge. Instead, most reports of strong overflow are from the part close to Iceland (Perkins et al., 1998; Voet, 2010; Olsen et al., 2016) and Beaird et al. (2013) estimate that the overflow across the northwesternmost part of the ridge is at least 0.5 Sv.

The northwesternmost passage across the IFR is the "Western Valley" (WV, Fig. 1) and the overflow through this passage, "WV-overflow", is the focus of this study. The WV is the deepest passage across this part of the IFR (Fig. 1) and there are at least two good arguments for expecting a strong and persistent WV-overflow.

The first argument is theoretical and based on analogy with other overflow sites. Most studies suggest that the deep and intermediate water in this part of the Nordic Seas generally has a cyclonic circulation (Nøst and Isachsen, 2003; Søiland et





al., 2008; Voet et al., 2010) as does the surface layer (Perkins et al., 1998; Jacobsen et al., 2003; Koszalka et al., 2011). The upstream source for overflow through the WV would then be located east of Iceland where dense water reaches close to the surface (e.g., Fig. 6 in Olsen et al., 2016). Simple overflow models that have been successful elsewhere (e.g., Whitehead, 1998) then imply a WV-overflow exceeding 1 Sv in volume transport. This overflow would be expected to flow along the

5 Icelandic slope and to cover the deep parts of the WV especially over its Icelandic flank. These assumptions are in agreement with the cold water of overflow character that is typically seen in this region on sections following the ridge crest as exemplified in Fig. 1b.

The second argument is based on moored current measurements downstream of the WV. Perkins et al. (1998) observed a strong (approx. 50 cm s$^{-1}$ core speed) and persistent bottom current at a site about 90 km downstream of the WV sill and

10 roughly 200 m deeper (blue circle labelled "P" in Fig. 1a), which they estimated to transport at least 0.7 Sv of "Arctic Intermediate Water", i.e. "pure overflow". The persistence of this current was confirmed by an ADCP, deployed for more than two years at almost the same location (Voet, 2010; Olsen et al., 2016). Beaird et al. (2013) have argued that this bottom current can only derive from overflow across the northwestern part of the IFR where the WV is the deepest passage.

Observations made in the WV itself are, however, more ambiguous than the evidence from the downstream

measurements. Thus, Perkins et al. (1998) also had moorings closer to the sill of the WV and they "find no evidence for significant flow through the WV". Likewise, Beaird et al. (2013) found that "the overflow transport in the WV is more variable than the current meter records of Perkins et al. (1998) suggest" where they presumably refer to the downstream measurements at site P.

This discrepancy was the main motivation for our field experiment, in which we placed three moorings along a section

across the WV close to the sill (red rectangle in Fig. 1a). The instrumentation was deployed in August 2016 and all the data were successfully recovered in May 2017. With the chosen configuration of measurements close to the bottom, we hoped to catch any overflow that might have bypassed the moorings of Perkins et al. (1998) in the WV. The experiment was also designed to test a low-cost system for long-term monitoring of the WV-overflow.

In the following, we first describe the materials and methods used and the results from the field experiment. In order to get

a longer term perspective, we compare our results with sea level variations measured by satellite altimetry and with historical hydrographic observations. From the results, we can derive estimates of the average volume transport of WV-overflow both for the duration of the field experiment and for longer periods and we discuss the validity of these estimates. Consistent with the results of Perkins et al. (1998), we find the WV-overflow to be much weaker than might be expected from the arguments presented above and we try to explain why that is the case using the results from a two-layer hydraulic model. Finally, we

discuss the implications of our results and present our conclusions and perspectives.

In order to keep the presentation more coherent, some of the details have been placed in the accompanying supplementary document, which also includes a more detailed description of the hydraulic two-layer model invoked in the discussion.





## 2 Material and methods

The main results in this study are based on a field experiment including the deployment of moored instrumentation at three locations in the WV (Table 1). To help interpret these data, we include also hydrographic observations from recent and historical CTD profiles in the region as well as sea level variations from satellite altimetry.

### 2.1 The field experiment 2016–2017

The three deployment sites were labelled A, B, and C (Fig. 3). At sites A and C, each package contained a SeaBird SBE39 temperature recorder and battery packs attached to a LinkQuest acoustic modem enclosed in a specially developed trawl-proof frame mounted on the bottom (Fig. S1, right panel, in supplementary document). Each of these "Bottom Temperature Loggers (BTLs)" recorded the bottom temperature at hourly intervals, which could be uploaded acoustically to a vessel.

At site B, a 150 kHz RDI Broadband ADCP and a SeaBird MicroCAT (SBE37) temperature and salinity logger were deployed within a trawl-proof frame (Fig. S1, left panel) that was attached by two acoustic releases to a concrete block mounted on the bottom. The MicroCAT recorded bottom temperature, conductivity, and pressure at 10 minute intervals. The ADCP recorded velocity at 30 levels (bins) with 10 m vertical resolution every 20 minutes.

The experimental design was based both on theoretical arguments and on historical hydrographic sections crossing the WV. With the chosen locations, we expected sites A and C to be close to the boundaries of the overflow plume passing through the valley. The measured bottom temperatures at these two sites would then document variations in these boundaries, i.e. the position of the plume in the valley. Likewise, the ADCP at site B was located where we expected the thickest overflow layer and the core of the plume (Fig. 4).

The two BTLs were placed permanently (no recovery option) at sites A and C and had battery capacities for many years. The intention was that the results from the field experiment, including the ADCP data, could be used to derive an algorithm giving overflow volume transport from the two bottom temperatures at sites A and C, alone. Since the BTLs may be interrogated from bypassing vessels, this could form the start of a low-cost monitoring system.

The instruments were deployed on August 14, 2016 by RV *Poseidon*. On May 20, 2017, RV *Magnus Heinason* uploaded the bottom temperature data from the BTLs at sites A and C and recovered the ADCP and MicroCAT at site B. During both cruises, CTD stations were occupied along a line crossing the valley (Fig. 3).

All temperature measurements from the deployed instruments were checked for data quality and appeared to be of high quality (Hansen et al., 2017a). The conductivity measurements of the MicroCAT showed spikes and a consistent drift through the observational period. They will not be considered here further. The temperature measurements at the three sites were then converted to hourly intervals and daily averaged bottom temperatures determined.

The ADCP data have been quality controlled using a semi-automatic routine, erroneous data flagged, and tidal current constituents determined (Hansen et al., 2017a). The tidal currents are quite strong over the IFR (Perkins et al., 1994) and this





is also seen in our ADCP measurements. Thus, the major semi-axis of the $M_2$ tidal current ellipse for the deepest bin (16.5 cm s$^{-1}$) was more than three times the magnitude of the average velocity of that bin (4.9 cm s$^{-1}$) (Table S1).

To generate time series of daily averaged velocity, the tidal velocity constituents, determined from the observations (Hansen et al., 2017a), were used to "predict" the tidal velocities for all bins during the field experiment. These were
5 subtracted from the original measurements and the resulting time series averaged to daily values for each bin. For the deepest bins, the de-tiding does not alter the daily averages appreciably but, for higher bins with large data gaps, it may prevent biasing in periods with strong tidal currents. The daily averaged velocities were of high quality up to bin 26, centred at 135 m depth, 267 m above the bottom (Table S1).

Both the deep and the upper level velocities were generally directed parallel to the valley axis (Fig. S2). We will use the
10 direction towards 225° as our x-axis and the velocity along this direction is denoted $U$ (Fig. 3). We will focus on the deepest measured velocity at 17 m above the bottom, which hereafter is denoted "deep velocity", $U_D$, and the uppermost high-quality velocity at 135 m depth, denoted "top velocity", $U_T$.

### 2.2 Hydrographic observations

In addition to the stations occupied during the deployment and recovery cruises, all CTD profiles within a rectangular area
(Fig. S3) around the WV (63.5° N to 65° N, 10° W to 14° W) acquired by the Faroe Marine Research Institute and the University of Hamburg were combined with CTD profiles from the NISE dataset (Nilsen et al., 2008) in the same area. This gave a total of 968 CTD profiles (Hansen et al., 2017b).

### 2.3 Altimetry data

Altimetry data were downloaded from the global gridded (0.25°×0.25°) data set available from Copernicus Marine
Environment Monitoring Service (CMEMS) (http://marine.copernicus.eu). We use the *Mean Dynamic Topography (MDT)* for the region covering the IFR (Fig. 2) and we use daily *Sea Level Anomalies (SLA)* for two altimetry grid points on either side of the valley (Fig. 3): point $h_1$ (64.375° N, 11.875° W) and point $h_2$ (64.625° N, 12.375° W). The difference in SLA value between these two points $\Delta h$ ($h_2 - h_1$) is a time series with daily values from January 1, 1993 to late June, 2017.

### 2.4 Statistical methods

For correlation analyses, we use standard linear (Pearson) correlation coefficients. To assess the statistical significance of the correlation coefficients, we use the "Modified Chelton Method" recommended by Pyper and Peterman (1998) to correct for serial correlation in the data. Significance level is indicated by asterisks attached to the correlation coefficient. * indicates $p < 0.05$, ** indicates $p < 0.01$, *** indicates $p < 0.001$. All of these are two-tailed probabilities. The uncertainty of regression coefficients is given as 95 % confidence intervals calculated using the degrees of freedom determined by the Modified
Chelton Method.



## 3 Results

### 3.1 In situ observations from the 2016–2017 field experiment

As shown by the detailed ETOPO1 bathymetry (Fig. 3), the moored instruments were deployed a short distance southwest of the sill of the WV, which appears to be at about 425 m depth. The region between sites A and C spans the deepest parts of the WV (Fig. 4) and any overflow through the WV would be expected to flow through this region with the core close to the ADCP at site B. An example of such a case is provided by the temperature distribution during the recovery cruise in May 2017 when cold water was covering the whole region between sites A and C (Fig. 4).

Typically (e.g., Dickson and Brown, 1994), the $\sigma_\theta > 27.8$ kg m$^{-3}$ criterion is used to define water sufficiently dense to be characterized as overflow. In our field experiment, we do not have sufficient salinity data to provide adequate coverage of density variation and are forced to use temperature instead. From CTD stations in the WV, it is seen that the temperature at which $\sigma_\theta = 27.8$ kg m$^{-3}$ occurs may vary considerably, but in about two thirds of the cases it is below 3 °C (Fig. S4). We will therefore use temperature less than 3 °C as our criterion and all water that fulfils this criterion will be called "Overflow Water" even though it may not necessarily cross the IFR into the Atlantic. We can then use the measured bottom temperatures to check the persistence of Overflow Water at the three sites (Fig. 5). The percentage of hourly observations with bottom temperature below 3 °C was 53 % for site A, 98 % for site B, and 75 % for site C.

Thus, site B was almost always covered by Overflow Water while site A was in Overflow Water half the time. Site A has therefore been at the average location of the northwestern boundary of the overflow layer, as was planned. Site C has been mainly within the overflow layer, but the bimodal temperature distribution indicates that it was sometimes covered by the warmer Atlantic Water. The average location of the southeastern overflow boundary has therefore been southeast of site C, but only a short distance to judge from the opposite slopes of isopycnals and bottom (Fig. 4). The distance between sites A and C is about 12 km and we therefore conclude that the average width of the overflow layer was less than 20 km during the field experiment.

The bottom temperature variation throughout the field experiment may include a seasonal signal at sites A and B, but apart from that, the variations seem unrelated (Fig. 6a). At site C, the bottom temperature changes rather abruptly between warm and cold conditions consistent with the bimodal bottom temperature distribution in Fig. 5. When de-trended to account for possible seasonal or long-term variations, weekly averaged bottom temperatures at the three sites were not significantly correlated with one another (Table 2).

Both top and deep velocities showed considerable variations during the field experiment (Fig. 6b). Averaged over the whole deployment period, the deep velocity was positive (i.e. towards the Atlantic), but it was weak, only 5 cm s$^{-1}$ (Table S1) and the along-valley velocity decreased with distance from the bottom (Fig. 7a, red line). Only a few profiles showed a "typical overflow shape" with a shallow overflow layer and a core some tens of meters above the bottom (Fig. 7b).

For a monitoring system based only on bottom temperature to be successful there has to be a significant relationship between bottom temperature and the velocity close to the bottom. The correlations between bottom temperature and deep





velocity (Table 2) are not, however, high. We do find (barely) significant negative correlations between $U_D$ and $T_B$ and between $U_D$ and $T_C$. Thus, a deep flow towards the Atlantic at site B is associated with colder water at these two sites, but the correlation coefficients are low and we find no correlation between $U_D$ and $T_A$. It is therefore not obvious how to monitor WV-overflow by the BTLs at sites A and C, solely.

Remarkably, the bottom temperatures at sites A and C are better correlated with the top velocity than with the deep velocity (Table 2). They are also significantly correlated with the vertical shear, which may be represented by $\Delta U = U_T - U_D$. This may be explained by the thermal wind equation that links shear and isoline tilt and has the implication that a strong Atlantic inflow ($U_T << 0$) is associated with increased influence of cold water at site A.

   Table 2 also shows a weak positive correlation between $U_D$ and $U_T$. This may be due to barotropic forcing from sea
level variations or may be caused by frictional drag between the upper and deep flows.

   In addition to the long section shown in Fig. 1b, RV *Poseidon* made a more detailed CTD survey of the WV during the deployment cruise in August 2016 (Fig. 8). On the northeasternmost section (I), upstream of the sill, the cold and dense water mass extended to fairly shallow levels. Closer to the sill (sections II and III), this water mass was much deeper. Downstream of the sill (sections IV and V), the cold water had descended further, but cold water was also seen on the
15 southeastern flank of the valley.

### 3.2 Altimetry

From geostrophy, we expect a linear relationship between the tilt of the sea surface from altimetry point $h_1$ to point $h_2$ (Fig. 3) and the surface velocity perpendicular to the line between the two points with the theoretical coefficient $\alpha_{Theoretical} = g/(f \cdot s)$ = 2.0 s$^{-1}$, where $g$ is gravity, $f$ the Coriolis parameter, and s i$s$ the distance between the altimetry points. Geostrophic balance
is verified by a high correlation coefficient of 0.86 between the SLA difference, $\Delta h$, and the top velocity, $U_T$, even though $U_T$ is measured at 135 m depth (Table 3).

   A linear regression analysis gave $U_T = \alpha_T \cdot \Delta h + \beta_T$ with $\alpha_T = (5.7 \pm 1.1)$ s$^{-1}$ and $\beta_T = (-14 \pm 3)$ cm s$^{-1}$. The value for $\beta_T$ is equivalent to a change in mean Dynamic Topography of -7 cm (-14/2.0) from altimetry point $h_1$ to point $h_2$, which is consistent with Fig. 2. The high value for $\alpha_T$ relative to $\alpha_{Theoretical}$ was to be expected since the surface current over site B is
25 probably much stronger than the horizontally averaged current between the two altimetry points (Fig. 3). Using the values of $\alpha_T$ and $\beta_T$ from the regression analysis, we can reproduce $U_T$ for the whole altimetry period (Fig. S5), which indicates that the Atlantic inflow branch over the WV was stronger than average ($U_T$ more negative) during our field experiment.

   The fact that $U_T$ and $U_D$ are significantly correlated (Table 2) indicates that $U_D$ might also be correlated with $\Delta h$. Table 3 lists a positive correlation coefficient between $U_D$ and $\Delta h$, but it is low and not significant. Similarly, we find low and non-
30 significant correlations between $\Delta h$ and the bottom temperatures at sites A and B, but the bottom temperature at site C is well correlated with $\Delta h$ as might be expected from the thermal wind equation and Table 2.





### 3.3 Hydrography close to the sill

From the CTD data base, including both recent and historical observations, we have selected stations in the sill region. As shown by the red arrows in Fig. 9a, we defined a coordinate system with origin at site B, x-axis towards 225° (Fig. 3), and y-axis towards 135°. We then selected CTD stations with $|y| \leq 50$ km and $|x| \leq 25$ km and plotted various parameters against the y-coordinate (Fig. 9b, c, d) although there is also considerable variation along the valley (x-axis) as demonstrated by the detailed CTD survey in Fig. 8.

The variation of bottom depth from the CTD stations along the y-axis (Fig. 9b) is consistent with a sill depth around 425 m, as indicated in Fig. 3. At most locations along this section, bottom temperature varies considerably, indicating shifts between overflow and Atlantic Water presence (Fig. 9c). Close to site B, however, the bottom water was consistently cold. In the interval from $y = -10$ km to $y = 6$ km (grey areas on Fig. 9b, c, d), there were altogether 39 stations with average bottom temperature 0.5 °C and none of them exceeded 3 °C. This is consistent with the observations from the field experiment that had very few hourly averaged bottom temperatures above 3 °C at site B (Fig. 5).

Using the 27.8 kg m$^{-3}$ isopycnal to denote the upper boundary of the overflow layer, a similar picture is seen (Fig. 9d). Along most of the section, there were CTD stations with the whole water column less dense than this value (isopycnal height = 0 in Fig. 9d), but not in the vicinity of site B.

From Table 3, one might perhaps expect that some of the variability in Fig. 9 could be derived from altimetry, but attempts to identify any such relationship did not give statistically significant results, whether considering bottom temperature or isopycnal depth/height.

### 3.4 Transport density

To get a first impression of the volume transport of overflow through the Western Valley, we integrate the average velocity profile towards 225° at site B (red curve in Fig. 7a) from the bottom up to the level where it is zero (about 60 m above the bottom). This gives a value of 1.5 m$^2$ s$^{-1}$, which is in the form of a transport density (volume transport per unit length across the valley).

This value should include both Overflow Water and any warmer water flowing towards the Atlantic and it might be considered to be an upper limit for the average transport density of Overflow Water at site B during the field experiment. Transport density might not, however, depend linearly on velocity, so this is not necessarily the appropriate average. In order to estimate a more reliable average and determine the variability, we have generated a time series that is intended to represent daily averaged transport density of Overflow Water at site B, which is termed "overflow transport density" and labelled $q_O$.

To do that requires daily estimates of the height of the overflow layer, which can be used as the upper integration limit of the velocity profile. As illustrated in Fig. 10, we use the 3 °C isotherm as the upper boundary and calculate its height above bottom at site B, $z$, for every day in two steps. The first step involves finding the height, $\Delta z_1$, of the 3 °C isotherm

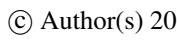



above (or below) bottom at site A by using the bottom temperature, $T_A$, at this site. The second step involves calculating the deepening (or rising), $\Delta z_2$, of this isotherm from site A to site B by using the tilt of the isotherm, $\phi$, which is assumed to be equal to the tilt of the 27.8 kg m$^{-3}$ isopycnal and may be derived by using the thermal wind equation:

$$\tan(\varphi) = \frac{\partial \rho / \partial y}{\partial \rho / \partial z} = \frac{\rho \cdot f \cdot (\partial U / \partial z)}{g \cdot (\partial \rho / \partial z)} \cong \frac{\rho \cdot f}{g} \cdot \langle (\partial \rho / \partial z)^{-1} \rangle \cdot \frac{\Delta U}{\Delta Z} \quad (1)$$

where $\rho$ is density, $f$ the Coriolis parameter, $g$ the acceleration of gravity, and instantaneous values of the inverse density gradient $(\partial \rho / \partial z)^{-1}$ have been replaced by their average value $\langle (\partial \rho / \partial z)^{-1} \rangle$. $\Delta U / \Delta Z$ is the vertical shear of the current determined from the ADCP profile over a vertical distance, $\Delta Z$, which was chosen to cover the depth interval over which the

10 isotherm deepens or rises from site A to site B.

An approximate value for $\langle (\partial \rho / \partial z)^{-1} \rangle$ was estimated from the CTD stations deeper than 300 m on Fig. 9a by averaging the vertical distance between the 27.75 and the 27.85 kg m$^{-3}$ isopycnals. These CTD data were also used to verify that the vertical distance between neighbouring isotherms for most cases is fairly constant (Fig. S6) and to find its average value, $\delta = 31$ m (Fig. 10). Both of these estimates as well as Eq. (1) are associated with considerable uncertainty and the

15 estimated values for $z$ will only be approximate but their validity may be evaluated independently by using the bottom temperature at site C.

This is done by extending the 3 °C isotherm further from site B to site C by a similar method, where $\Delta U / \Delta Z$ is calculated from the bottom up to the estimated isotherm height at site B. The correlation coefficient between the calculated isotherm height and bottom temperature at site C was -0.73*** for weekly averages. This adds confidence to the procedure

for calculating $z$ and we have therefore used the estimated values for $z$ (Fig. S7) to generate a time series for $q_0$ by integrating the ADCP profile from the bottom up to $z$.

Averaged over the duration of the field experiment, the overflow transport density, $q_0$, was 1.4 m$^2$ s$^{-1}$ and its variations were related to variations of the top velocity, $U_T$, with a highly significant correlation coefficient $R = 0.51$*** for daily averages, which increased to $R = 0.64$*** for weekly averages (Fig. 11a).

Since the weekly averaged values for $U_T$ were highly correlated with the SLA difference $\Delta h$ (Table 3), this link between $q_0$ and $U_T$ indicates that there may also be a link between $q_0$ and $\Delta h$. That is indeed the case since the correlation coefficient between $q_0$ and $\Delta h$ was $R = 0.60$** for weekly averages (Fig. 11b).

Although the correlation coefficient in Fig. 11b is significant, the relationship is not necessarily linear. Nevertheless, the scatter seems too high to warrant high-order fits. We have therefore used the regression coefficients of Fig. 11b to

30 reproduce values for $q_0$ for the whole altimetry period (Fig. 12). The seasonal variation of $q_O$ does not seem to be very pronounced but there is an indication of strengthened overflow during summer (Fig. 12b). The long-term variation shows that our field experiment is likely to have been during a period of weak WV-overflow (Fig. 12a). From the beginning of



1993 to the end of 2016, the regression equation gives an average value of 2.7 m$^2$ s$^{-1}$ for $q_0$. This long-term estimate is almost twice as high as during the field experiment.

In spite of the apparent consistency between the estimated values for $z$ and the bottom temperature at site C, there is still considerable uncertainty associated with $z$ and hence also $q_0$. Also, the choice of the 3 °C isotherm as the upper boundary of the overflow layer introduces some uncertainty (Fig. S4). The effects of both may be estimated by running the analysis with different choices for the isotherm used as the upper boundary (Table 4). Going from 3 °C to 4 °C (by adding δ = 31 m to $z$ for every day) increased the value of $q_0$ only slightly for the field experiment, but some more for the whole altimetry period. Going to higher isotherms actually reduced the estimates of $q_0$. This indicates that the maximum estimate of the long-term value for $q_0$ is more constrained than might be thought from the high uncertainties.

## 4 Discussion

### 4.1 Average volume transport of WV-overflow

The results of our field experiment present us with a conundrum: On the one hand, water with overflow character is present at site B almost all the time and also shows little variability in historical hydrographic data. Since the site is close to the sill, this water would be expected to flow more or less continuously towards the Atlantic. Instead, the daily averaged deep velocity is away from the Atlantic 30 % of the time and even when it is towards the Atlantic, it is weak and variable (Fig. 6b). Only a few days showed typical overflow velocity profiles during the field experiment (Fig. 7b).

In addition to this, there are the arguments for a strong and persistent WV-overflow based on theory and observations that were mentioned in the introduction. We therefore need to ask whether the instruments deployed in the field experiment were located so that they missed the overflow plume.

That is not likely. The bottom temperature distributions (Fig. 5) combined with bottom topography (Fig. 4) indicate that site A and site C are close to the upper and deeper average boundaries of the overflow layer, respectively. Bottom temperatures from the field experiment (Fig. 6a) and from the historical CTD data set (Fig. 9c) likewise confirm that the coldest bottom water is usually found close to site B as are the largest isopycnal heights (Fig. 9d).

Thus, the average velocity of the overflow core is not likely to be much stronger than the ADCP measurements at site B (Fig. 7) while the bottom topography (Fig. 4) combined with the expected interface slope indicate that the height of the overflow layer also is close to its maximum at site B. The average transport density measured at site B (1.4 m$^2$ s$^{-1}$) is therefore likely an overestimate for the overflow layer as a whole. In Sect. 3.1, we argued that the average width of the overflow layer during the field experiment was less than 20 km. We therefore conclude that the transport of Overflow Water (colder than 3 °C) through the WV was less than (1.4 m$^2$ s$^{-1}$)×(20 km) ≈ 0.03 Sv on average during the field experiment.

These arguments are supported by the conclusions from the inviscid two-layer hydraulic model in the supplementary document, which indicates that a strong WV-overflow should have been seen by the ADCP at site B. The model also verifies





that the overflow transport density at site B multiplied by a fixed width of 20 km generally will give an overestimate of the volume transport.

The overflow transport density, $q_O$, has both positive and negative values, which reduces the average to a small value. One might be tempted to argue that only the positive values should be used, but that would only be appropriate if we can

assume that most of the water flowing towards the Atlantic during positive transport events would continue in that direction and that the return flow (towards the Norwegian Sea) outside of the events would carry a different water mass; not Overflow Water. The negative correlation coefficient between $T_B$ and $U_D$ in Table 2, does indicate that the returning water ($U_D$ negative) tends to be warmer than during overflow events ($U_D$ positive), but the difference is not very pronounced and the returning water was always sufficiently cold to be classified as overflow (Fig. 5 and Fig. S8).

Thus, it seems difficult to argue that the average overflow transport during the field experiment was higher than 0.03 Sv. From Fig. 12a it appears, however, that $q_O$ and hence also WV-overflow was relatively weak during our field experiment and that the average of $q_O$ over the whole altimetry period was twice as high. Uncertainties in the regression coefficients (Fig. 11b) imply an uncertainty in $q_O$ of about 2 m$^2$ s$^{-1}$. In addition to this, there is the uncertainty in our estimate of the height of the 3 °C isotherm, $z$, and using a different criterion for defining Overflow Water will also affect $q_O$. Both of these

were estimated in Table 4, which indicates that they at most can increase the long-term estimate of $q_O$ by less than 1 m$^2$ s$^{-1}$.

A maximum estimate of the long-term average $q_O$ is therefore around 5 m$^2$ s$^{-1}$, which implies a maximum long-term average WV-overflow transport of (5 m$^2$ s$^{-1}$)×(20 km) = 0.1 Sv. This value represents the maximum average transport of Overflow Water i.e., water colder than 3 °C, but even including water up to 6 °C in temperature would not increase this value according to Table 4.

**4.2 Why is WV-overflow so weak?**

From our field experiment, we thus find an overflow volume transport that is much less than what one might expect from the arguments presented in the introduction. The problem is conveniently illustrated by two of the sections in Fig. 8. On section I, the 27.8 kg m$^{-3}$ isopycnal reaches almost to the surface, but on the neighbouring section II, it is close to the bottom. Why does the cold and dense water on section I not penetrate to section II and further through the WV?

The answer seems to be associated with the Atlantic inflow to the Norwegian Sea in the upper layers over the WV. From Fig. 11a, a strong Atlantic inflow is associated with a weak or negative overflow transport density and there are several possible explanations for this:

- Frictional drag between the Atlantic inflow and overflow will tend to reduce overflow velocity.
- The vertical shear between the Atlantic inflow and the overflow will make the interface between the two water
masses tilt more strongly, making the overflow layer both thinner and narrower as the strong correlation between $T_C$ and $\Delta U$ in Table 2 verifies.
- The flow path of the Atlantic inflow is associated with a sea level drop, $\Delta \eta$, from the WV to the region east of Iceland (Fig. 2) as illustrated in the idealized sketch in Fig. 13. The barotropic forcing induced by this ($g \cdot \rho_0 \cdot \Delta \eta$) will





counteract the baroclinic forcing that drives the overflow ($g \cdot \Delta\rho \cdot \Delta D$). With a density difference $\Delta\rho = 0.5$ kg m$^{-3}$, a sea level difference, $\Delta\eta$, of 10 cm can neutralize a difference in interface depth, $\Delta D$, of 200 m (Fig. 13).

The last two points above are elaborated in the inviscid two-layer model (Fig. S9), which confirms that with realistic choices of parameters, these effects can lead to a very weak WV-overflow. The model also demonstrates the effect of the bottom topography. A weakly sloping bottom will lead to a thin overflow layer and a weak transport.

Even without friction, the effects of the Atlantic inflow are thus sufficient to reduce the WV-overflow to the level that we have observed and friction will reduce the overflow even more. How important frictional drag is compared to the inviscid mechanisms listed above is difficult to assess but taken together, all these mechanisms are clearly able to allow the Atlantic inflow to suppress the WV-overflow to the level observed.

## 4.3 Implications

If WV-overflow is as weak as our observations indicate, how can we explain the persistent bottom current with average core speed of 50 cm s$^{-1}$ at site P some 90 km downstream from the WV sill (Fig. 1)? As argued by Perkins et al. (1998) and Beaird et al. (2013), the original source of this current must be overflow across the IFR. The width of the current and hence its volume transport are, however, not well constrained by observations and neither is the transport increase due to entrainment downstream of the ridge. Nevertheless, it has been argued by Perkins et al. (1998) that the original overflow feeding this current must have been at least 0.7 Sv whereas Voet (2010) suggested an average of 0.5±0.3 Sv.

A reliable estimate of the volume transport of the bottom current at site P and its source will probably require additional observational effort. Nevertheless, it seems doubtful that WV-overflow can be its only source even though Fig. 12a indicates that WV-overflow was considerably stronger during the ADCP measurements at site P (Voet, 2010) than during our field experiment. That still leaves other parts of the IFR, however, and Beaird et al. (2013) argued that overflow from the whole northwestern half of the IFR could feed into this current. This is supported by the presence of cold bottom water south of the WV seen in historical CTD profiles (Fig. S3).

In their map summarizing the currents southeast of Iceland (their plate 1), Perkins et al. (1998) show a bottom flow circulating into and out of the Atlantic end of the WV. Such a current (curved blue arrow on map of Fig. 8) would explain the cold water over the southeastern slope of the valley on sections IV and V of Fig. 8 and would be a pathway for Overflow Water from more southerly parts of the ridge to flow towards site P.

This current may be fed by two different mechanisms generating Overflow Water across the IFR. One is the mechanism illustrated in Fig. 13, by which cold, dense water of Arctic origin pushes its way across the ridge analogous to the overflows through the Denmark Strait and Faroe Bank Channel. This mechanism is likely strongest in summer (Voet, 2010; Olsen et al., 2016) and is the mechanism represented in hydraulic two-layer models (e.g., Wilkenskjeld and Quadfasel, 2005; Voet, 2010; Olsen et al., 2016) like the one presented in the supplementary document (Fig. S9).





The other mechanism is subduction along the Iceland-Faroe Front (Read and Pollard, 1992; Perkins et al., 1998), which is strongly winter-intensified (Beaird et al., 2016). The resulting water mass has a low salinity and most of it is not sufficiently dense to fulfil our criterion for Overflow Water. Part of it joins the Atlantic inflow to the Norwegian Sea (Beaird et al., 2016), but any fraction that flows into the Atlantic is likely to join the bottom current at site P. Our rejection of the

Western Valley as an important overflow contribution is therefore not incompatible with the observations at this site.

When considering their contribution to the North Atlantic Deep Water and AMOC, the Overflow Waters generated by these two mechanisms play similar roles although modified by density differences. For budget estimates of the Arctic Mediterranean, in contrast, they behave quite differently since the frontal subduction of Atlantic Water occurs before the Atlantic Water reaches the section where volume transport of Atlantic inflow between Iceland and Faroes is monitored

(Hansen et al., 2015). Overflow Water generated by frontal subduction therefore should not be included in these budgets.

## 5. Conclusions and perspectives

From the measurements of our field experiment we estimate the average WV-overflow during the experiment to have been less than 0.03 Sv. Our experiment was probably during a period of relatively weak WV-overflow but combining these measurements with historical CTD observations and satellite altimetry, we conclude that the volume transport of overflow

through the WV has been at most 0.1 Sv on average over the last two decades.

The main reason for this low value seems to be the Atlantic Water flowing into the Norwegian Sea in the upper layers, which may suppress the overflow in at least three different ways. Firstly, friction between the oppositely moving Atlantic inflow and overflow may slow down the overflow. Secondly, the shear between the Overflow Water and the Atlantic Water is associated with increased tilt of the interface between the two water masses so that the overflow layer becomes thinner and

narrower and its volume transport weaker. Thirdly, the Atlantic inflow is associated with a sea level difference between the WV sill region and the upstream basin feeding the overflow. The barotropic pressure gradient induced by this sea level difference counteracts the baroclinic pressure gradient at depth, which otherwise would push the Overflow Water through the WV.

All of these tend to reduce the overflow and make the interface deeper. The effect of a deeper interface is further

enhanced by a weakly sloping bottom on the Icelandic flank of the WV, which also reduces the height of the overflow layer and allows bottom friction to prevent high overflow velocities.

With our determination of a weak WV-overflow, one piece of the IFR-overflow puzzle has been found, but the whole picture still seems rather hazy. A future, more reliable, estimate of the volume transport of the bottom current at site P (Fig. 1a) might help considerably. For the present, we can only conclude that the total IFR-overflow is likely to be considerably

less than the canonical 1 Sv value.

One of the motivations for the field experiment was to design a monitoring system for WV-overflow based on the bottom temperature loggers at site A and site C. The results from the field experiment did not support the basic hypothesis on



which this system was based and indicate that monitoring might be better achieved through the use of altimetry data. At the same time, the low transport value makes the motivation for monitoring WV-overflow rather doubtful.

**Data availability**

The observational data from the moored instruments during the field experiment as well as the CTD data acquired by the
Faroe Marine Research Institute are available at www.envofar.fo while the CTD data acquired by the University of Hamburg during the deployment cruise are available at https://pangaea.de/.

**Competing interests**

The authors declare that they have no conflict of interest.

*Acknowledgements.* The 2016–2017 field experiment as well as part of the analysis was funded by the Danish Energy Agency as part of the Arctic Climate Support Programme (Western Valley Overflow project). CTD data was obtained through the Co-Operative Project ''RACE II – Regional Atlantic Circulation and Global Change'' funded by the German Federal Ministry for Education and Research (BMBF), Förderkennzeichen 03F0729B. This study was supported by the Blue-Action project, which has received funding from the European Union's Horizon 2020 research and innovation
programme under grant agreement No 727852.

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





**Table 1.** Positions and bottom depths of the three deployment sites. BTL indicates Bottom Temperature Logger.

| Site | Instruments | Latitude | Longitude | Depth |
|------|-------------|----------|-----------|-------|
| A | BTL | 64.477° N | 12.139° W | 297 m |
| B | ADCP+MicroCAT | 64.445° N | 12.063° W | 402 m |
| C | BTL | 64.401° N | 11.967° W | 433 m |



**Table 2.** Correlations between various combinations of weekly averaged observed parameters with statistical significance indicated. These are bottom temperature ($T_A$, $T_B$, $T_C$) and along-valley components (towards 225°) of deep velocity ($U_D$) and top velocity ($U_T$). $\Delta U$ is the difference between top and deep velocity ($\Delta U = U_T - U_D$). Only statistically significant correlation coefficients ($p < 0.05$) are shown. The others are listed as "n.s.". All parameters have been linearly de-trended before correlation. Introducing lags between the parameters did not
5 give better correlations for any of the cases in the table.

|       | $T_B$ | $T_C$ | $U_D$ | $U_T$ | $\Delta U$ |
|-------|-------|-------|-------|-------|-----------|
| $T_A$ | n.s.  | n.s.  | n.s.  | 0.38* | 0.38*     |
| $T_B$ |       | n.s.  | -0.39* | n.s. | n.s.      |
| $T_C$ |       |       | -0.41* | -0.76*** | -0.66*** |
| $U_D$ |       |       |       | 0.42* | n.s.      |
| $U_T$ |       |       |       |       | 0.92***   |





**Table 3.** Correlation coefficients between weekly averaged in situ parameters and the difference, $\Delta h$, of SLA values between the two selected altimetry grid points (Fig. 3). All parameters have been linearly de-trended before correlation.

| $U_T$ | $U_D$ | $T_B$ | $T_A$ | $T_C$ |
|---|---|---|---|---|
| 0.86*** | 0.37 | -0.08 | 0.25 | -0.76*** |



**Table 4.** Estimated average values for the overflow transport density at site B, $q_0$, during the field experiment and during the altimetry period for different choices of the isotherm defined as the top of the overflow layer.

| Isotherm: | 2 °C | 3 °C | 4 °C | 5 °C | 6 °C |
|---|---|---|---|---|---|
| Field exp. (m² s⁻¹): | 1.3 | 1.4 | 1.5 | 0.7 | -1.5 |
| Altim. per. (m² s⁻¹): | 1.9 | 2.7 | 3.5 | 3.4 | 2.2 |





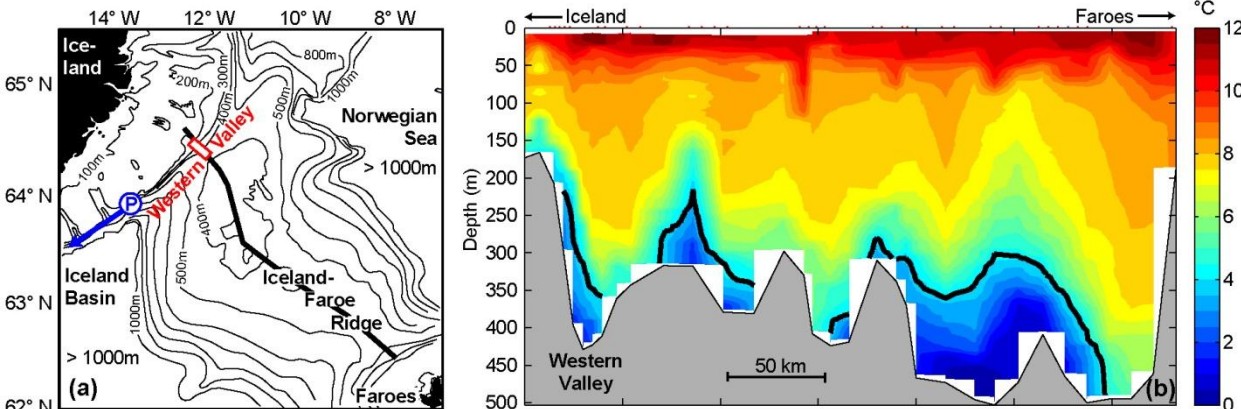

**Figure 1. (a)** Bottom topography of the Iceland-Faroe Ridge with isobaths for every 100 m down to 1000 m depth. The red rectangle in the Western Valley indicates the instrument deployment sites of our field experiment. The blue circle marks site P and the blue arrow indicates the strong and persistent bottom current at this site reported in previous studies. The thick black line along the ridge crest indicates a CTD section occupied by RV *Poseidon* in August 2016 and the potential temperature distribution along that section is shown as background colour in **(b)** with the black line indicating the $\sigma_0 = 27.8$ kg m$^{-3}$ isopycnal, commonly used as the upper boundary of the overflow layer.





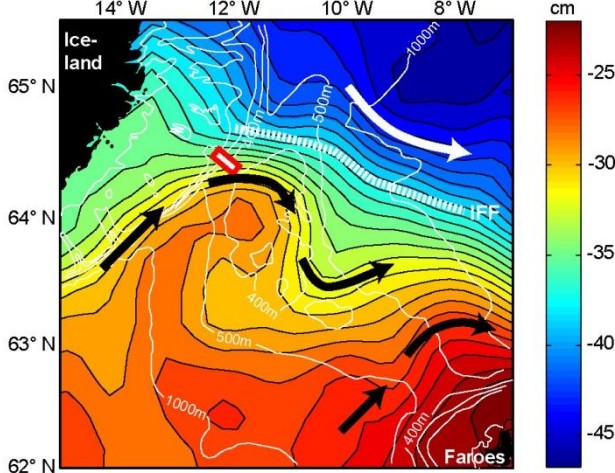

**Figure 2.** The Mean Dynamic Topography (background colours) above the IFR according to altimetry data from Copernicus Marine Environment Monitoring Service (CMEMS) (Sect. 2.3). Isobaths are shown by thin white lines for every 100 m down to 500 m and for 1000 m depth. Black arrows indicate the two main Atlantic inflow regions over the IFR. The white arrow indicates the surface circulation 5 of the Norwegian Sea. The thick hatched white line indicates the Iceland-Faroe Front (IFF). The red rectangle marks the deployment sites of our field experiment.





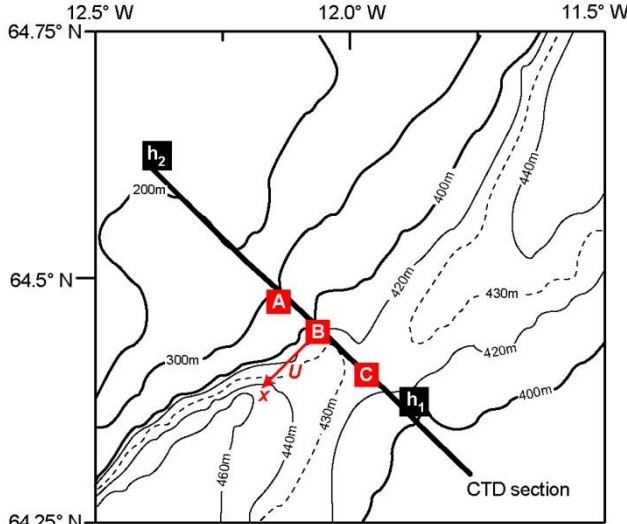

**Figure 3.** The region around the Western Valley with bottom topography based on ETOPO1. The positions of the three moorings of the field experiment are marked by red squares. Black squares indicate the two selected altimetry grid points. The thick black line shows a
5    CTD section that was occupied by RV *Poseidon* during the deployment cruise in August 2016 and by RV *Magnus Heinason* during the recovery cruise in May 2017. The red arrow indicates the chosen x-direction for along-valley flow towards the Atlantic (225°) with velocity component labelled *U*.




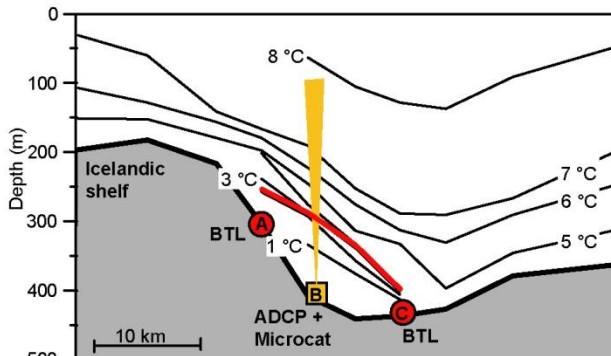

**Figure 4.** Section through the three moorings of the field experiment (Fig. 3, thick black line). The yellow cone indicates the maximal ADCP range. The isotherms show the temperature distribution on May 20, 2017 as observed by RV *Magnus Heinason*. The red line shows the $\sigma_\theta = 27.8$ kg m$^{-3}$ isopycnal.





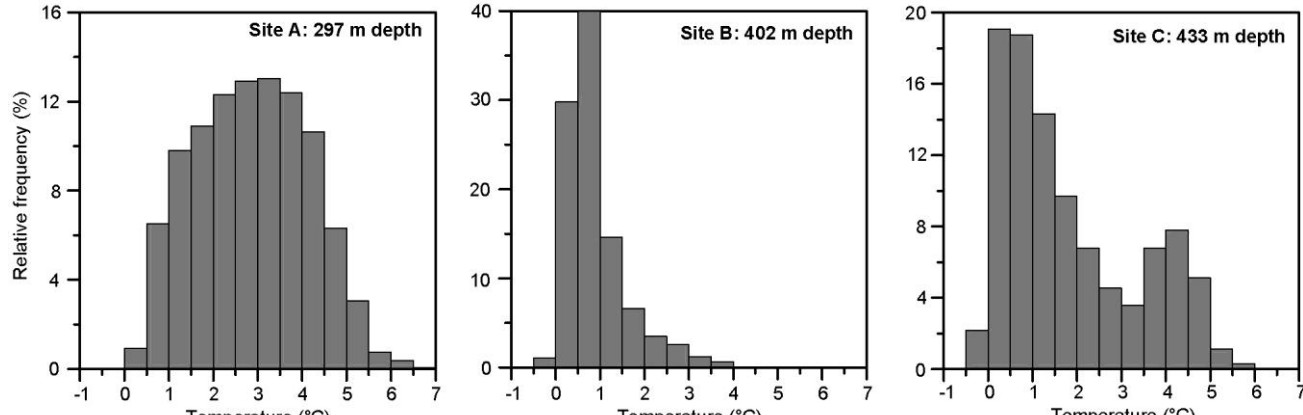

**Figure 5.** Histograms of bottom temperature at the three deployment sites based on hourly values.




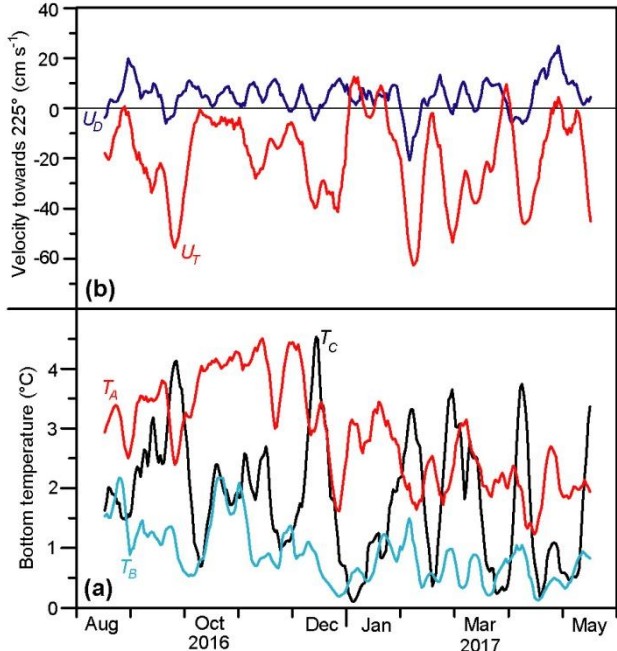

**Figure 6. (a)** Weekly averaged (7 day running mean) bottom temperatures at the three sites. **(b)** Weekly averaged velocity towards 225° at two depths at site B: deep velocity, $U_D$, at 385 m depth and top velocity, $U_T$, at 135 m depth.





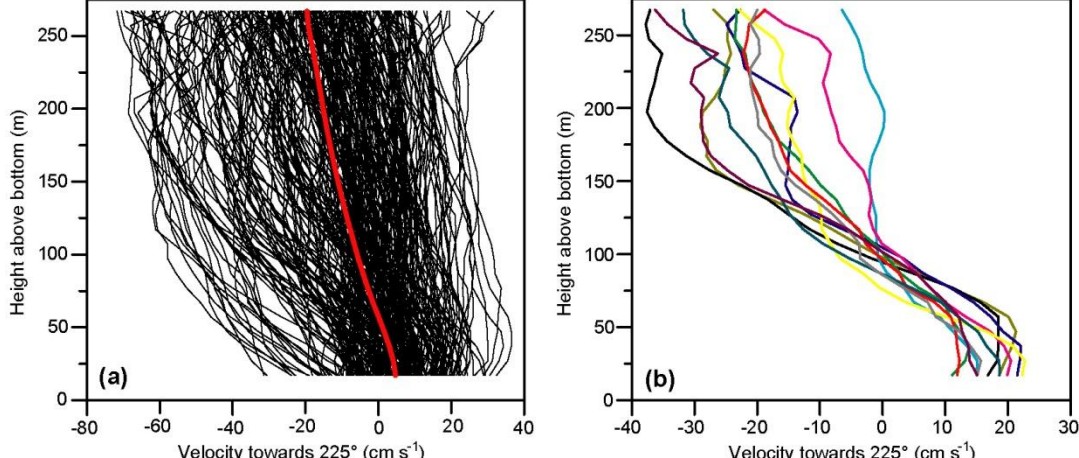

**Figure 7.** Vertical profiles of $U$ as observed by the ADCP at site B on every day **(a)** and on those days, for which the velocity 47 m above the bottom (bin 4) was $\geq 10$ cm s$^{-1}$ while the velocity 107 m above the bottom (bin 10) was $\leq 0$ **(b).** Black lines on **(a)** indicate individual daily averaged profiles while the thick red line indicates the average profile for the whole deployment period.





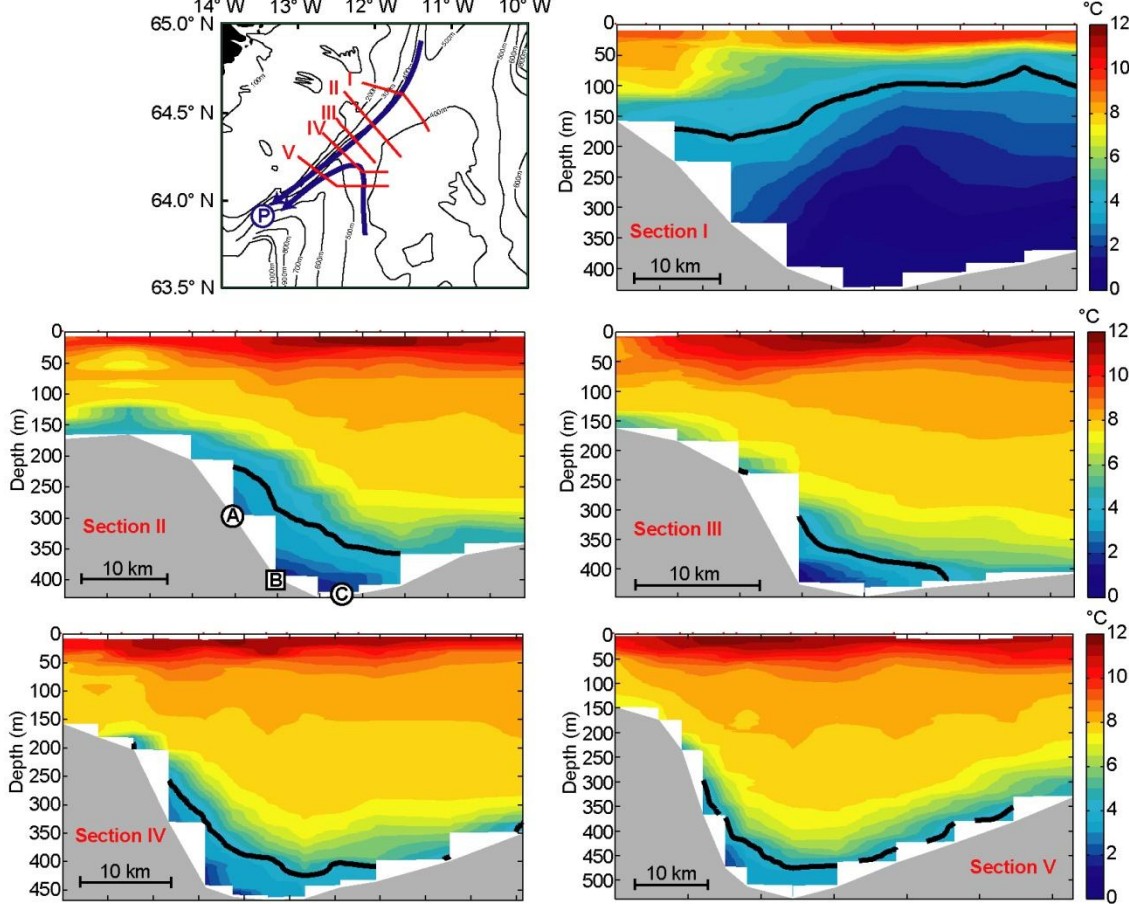

**Figure 8.** Station map and potential temperature (background colour) along five sections (I to V) crossing the WV, acquired by RV *Poseidon* in August 2016. The thick black lines indicate the $\sigma_\theta = 27.8$ kg m$^{-3}$ isopycnal. Vertical and horizontal scales vary between sections, but the temperature (colour) scales are identical. The moorings of the field experiment were deployed along section II and are shown at the bottom of that section. The blue arrows on the map indicate likely cold-water flow paths and the blue circle labelled P on the map marks mooring site P.





**Figure 9. (a)** Selected CTD stations in the sill region with coordinate system indicated by red arrows. **(b, c, d)** Variation along the y-axis of bottom depth **(b)**, bottom temperature **(c)**, and height of the $\sigma_\theta = 27.8$ kg m$^{-3}$ isopycnal above bottom **(d)**. Isopycnal height = 0 indicates that the whole water column was less dense than 27.8 kg m$^{-3}$. Bottom temperature is shown only for profiles with the CTD reaching less than 40 m from bottom. The origin of the y-axis is located at site B as indicated by the vertical line on **(b, c, d)** and the grey areas highlight the region from $y = -10$ km to $y = 6$ km.

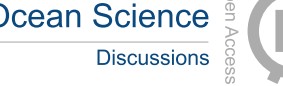

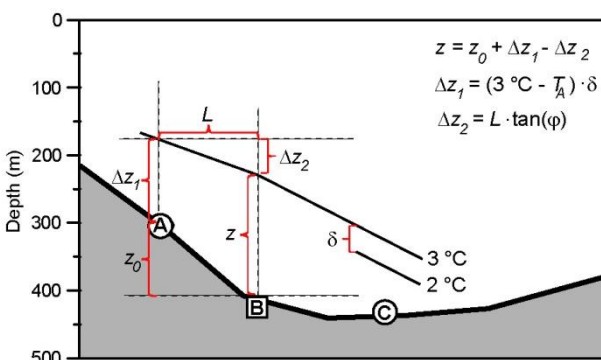

**Figure 10.** Sketch illustrating the determination of the height of the overflow layer at site B, $z$. If daily averaged bottom temperature at site B was $> 3$ °C, $z$ was set to zero. Otherwise, $z$ was required not to be less than a minimum value of 25 m.





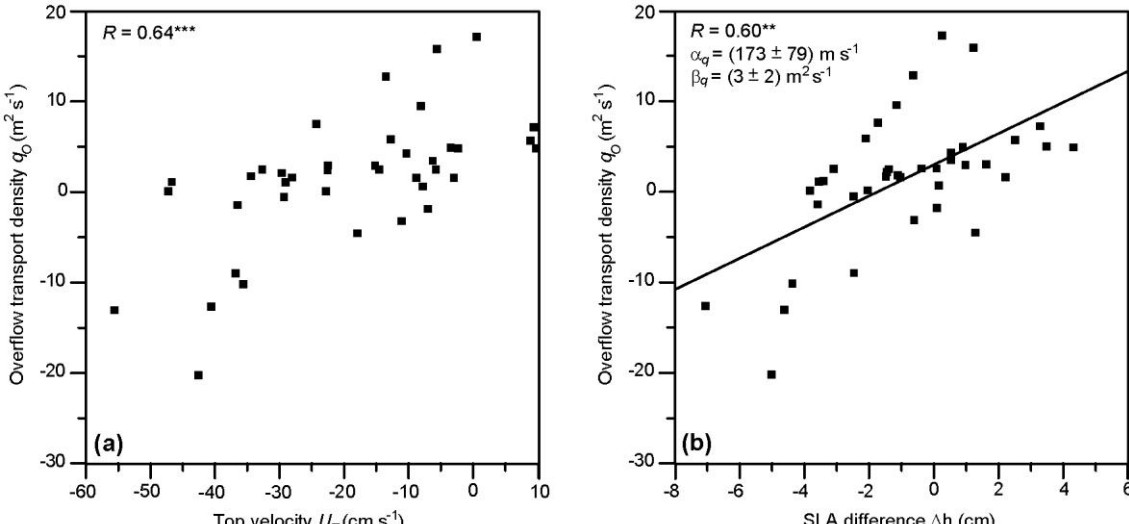

**Figure 11.** Weekly averaged overflow transport density at site B, $q_O$, plotted against weekly averaged top velocity $U_T$ **(a)** and against weekly averaged difference in SLA values $\Delta h$ **(b)**. The correlation coefficients ($R$) are shown with statistical significance and in **(b)** also the regression coefficients and the regression line: $q_0 = \alpha_q \cdot \Delta h + \beta_q$.




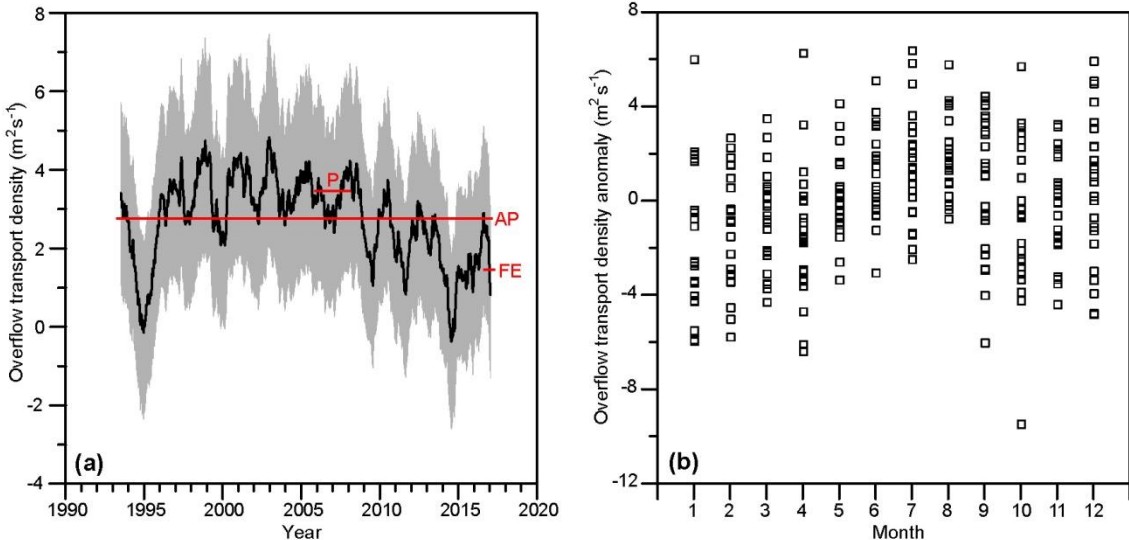

**Figure 12. (a)** Low-passed (365 day running mean) overflow transport density at site B, $q_O$ (black curve), with uncertainty interval (grey area) from 1993 to 2016 based on satellite altimetry and the regression equation in Fig. 11b. Red lines indicate the estimated overflow transport density averaged over the field experiment (FE), the whole altimetry period (AP), and the duration of the ADCP measurements at site P (P). **(b)** Monthly averages of overflow transport density anomaly, defined as the deviation from the 365 day running mean centred on the month.



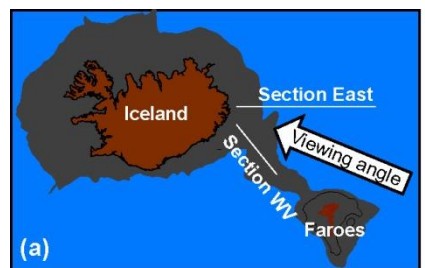
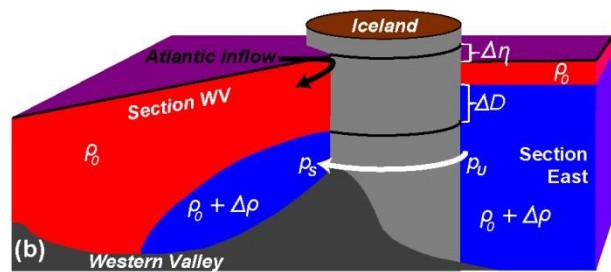

**Figure 13.** An idealized sketch of the region looking from the Norwegian Sea towards Iceland (viewing angle in **a**) showing two sections: one going eastwards (Section East) and one crossing the WV (Section WV). **(b)** The ocean is approximated as a two-layer system with density difference $\Delta\rho$. The Atlantic inflow turns southwards after passing through the WV, which is associated with a sea level difference $\Delta\eta$ between the two sections (Fig. 2). The thick white arrow illustrates a water parcel of dense water flowing along the Icelandic slope from east of Iceland through the WV without changing depth. Ignoring friction, the Bernoulli equation gives the acceleration of the water parcel in terms of the pressure change along the path: $\Delta p = p_U - p_S = g \cdot (\Delta\rho \cdot \Delta D - \rho_0 \cdot \Delta\eta)$.