# Peer review of "Overflow of cold water across the Iceland-Faroe Ridge through the Western Valley"

_Ocean Science, 2018_

## Referee Comment (RC1) · J. Whitehead (Referee) · 15 May 2018

This is a fine study in great detail of a flow that really needed to be measured. The measurements are well done, the discussion is excellent and the figures and theory are clearly presented. I have no negative comments but one suggestion.

In the introduction there should be a showing more clearly where the study area is compared to the other two overflows. For a bit of time, I was confused.

---

## Referee Comment (RC2) · Anonymous Referee #2 · 20 May 2018

This manuscript is very interesting and stunning. It all verses on how weak the overflow on the Iceland-Faroe Ridge is and tries to find a logical explanation for it. To fulfill the longtime series, in situ hydrographic observations are combined with altimetry data. The paper concludes that even thought the measurements took place on a low overflow period, in average low transports values should be expected. Another important conclusion arises from a model, the inflow of Atlantic Water is able to suppress the overflow. Even though I found this work interesting and easy to follow, I missed having error estimations and I also encountered with figures with bad caption or even turn around. Thus, my recommendation for this paper is to publish it after some changes. Beneath a list of the comments I have: Mayor Comments: - Section 2.2. We observe in one of the plots where each CTD section took place. However, it would be interesting

to add a line here about what distance exists in between the profiles or a map of their location. - Supplemental Figure 4 is used to stablish an important criterion to define overflow water on the manuscript. Thus, I think it should be included in the manuscript and not in supplementary material. It also highlights the distribution of overflow water. - Page 6 line 11. Maybe here it can be stated that even though throughout the manuscript the 3°C isotherm is used as upper limit for the overflow water, at he end of section 3 a sensitivity analysis is carried out. - Figure 6 needs to be turn upside down so the upper side is Bottom temperature and the lower velocities to be consistent with the text and caption. - The caption of Figure 7 is also opposite to the figure and text. - Figure 8. Could you also add a line for the 3°C on the sections to compare with the 27.8 kg m-3 isopycnal. Could the map be slightly bigger so one can read the isobaths? - Using the terminology of transport density when part of your data has density and part not, even if it has nothing to do, creates confusion. Better use the terminology of (volume) transport per unit length. - Page 8 line 22 and Page 10, in lines 26 and 29, please add the uncertainty that these values have. - Table 4 add uncertainty.

Minor Comments: - I think that breaking the author breaks the flow of the reading by trying to have small paragraphs. I think the following paragraphs pairs can benefit from blending into one: (1) starting on page 2 from line 30, to page 3 in line 7, (2) page 3 starting on line14 and ending in line 23, (3) page 3 starting on line 24 and ending in line 32, (4) page 10 from line 17 to line 23 - Page 6, Line 4 remove: "which appears to be", is it or is it not? - Figure 5. Please advise the reader that each figure has a different y axis on the caption.

---

## Referee Comment (RC3) · Anonymous Referee #3 · 5 Jun 2018

Review: Overflow of cold water across the Iceland-Faroe Ridge through the Western Valley, Hansen et al., Ocean Science Discussions

Summary:

Hansen et al. present current meter records from the Western Valley of the Iceland-Faroe Ridge which is the deepest channel in the north-western part of the ridge. However, this paper shows a much smaller flux than expected (max 0.1 Sv). Using the relationship between the instruments and SSH, the authors reconstruct the transport through the WV since 1993, showing that the flux during the field campaign was particularly low. However, even using the long-term mean transport, overflow through the WV is insufficient to explain the previously reported transport downstream of the ridge.

[Figure]

The paper is interesting and scientifically robust. The authors have been very thorough and I have very few comments as I found the authors have already answered them all in the manuscript. The findings add to the knowledge of overflow over the Iceland-Faroes Ridge and I think this study will spawn further research. Additionally, the paper is well written and the figures are of a high quality. As such I recommend acceptance after very minor revisions.

Comments:

Lines 14-19, p.6: I think that the different distributions at A and C suggest the two sites possibly have different mechanisms going on. A seems to be almost normally distributed around about 3°C – sometimes it has colder water, sometimes warmer. I wondered if it was as a result of a change in the lateral or vertical position of the overflow interface depending on the volume of overflow (or other effects). In contrast C has a bi-modal structure – it is either overflow water, or water centred upon 4°C. I wondered whether this site is mostly within the overflow, but that it is sometimes replaced by (northward flowing?) Atlantic water?

Lines 5-6 p.7: suggest addition that U(T) and T(A) are positively correlated, whilst U(T) and T(B) are negatively correlated.

Lines 1-8, p.7: think you need to add a sentence to remind the readers the velocity measurements are at B only. The velocity could change quite a lot laterally? I think I find it most surprising that U(D) and T(B) are not significantly correlated!

Lines 12-13 p.11: could you add an additional sentence or two explaining how you calculated the uncertainties associated with the regression coefficients?

Lines 18-23, p.12: your figure 1b also suggests there may be other places along the IFR where overflow is stronger than through the WV.

I found some of the figure captions difficult to follow. For some figures (e.g. 6) you had: (a) description, (b) description; whilst for others (e.g. 7) you have: description (a),

description (b). I think sticking to one format would help the reader.

Figure 4: could you define BTL in figure caption (for those of us who look at the figures before reading the article)

Figure 4: suggest addition of where the CTD stations were (e.g. by crosses on upper x-axis)

Figure 5: either plot all on same y-axis, or put note to reader in caption that different y-axis are used.

Figure 6: swop round so (a) is top panel, and (b) bottom panel

Figure 9: could you add the locations of A and C onto the plots too? I think this would be interesting to see

Table 1: were all the instruments 17 m off bottom? (as mentioned in the text)

---

## Author Comment (AC2) · 25 Jul 2018

**Responses to referee comments on "Overflow of cold water across the Iceland-Faroe Ridge through the Western Valley" by Bogi Hansen et al.**

We thank J. Whitehead and two anonymous referees for inspiring and constructive comments. Below, we address all the comments and describe our responses to them where we refer to the revised manuscript by listing the page and line numbers of changes in the format [page , line ]. In the revised manuscript, all changes from the original text are coloured red, except for changed figure numbers, which are due to the addition of two new figures requested by the referees.

**Referee J. Whitehead**

Comment 1.1: In the introduction there should be a showing more clearly where the study area is compared to the other two overflows. For a bit of time, I was confused.
Response: A new Figure 1 has been added.

**Anonymous Referee #2**

Comment 2.1: Section 2.2. We observe in one of the plots where each CTD section took place. However, it would be interesting to add a line here about what distance exists in between the profiles or a map of their location.
Response: The text has been modified so that it more clearly refers to a map of profile locations (Fig. S3). [page 5, line 16-17].

Comment 2.2: Supplemental Figure 4 is used to stablish an important criterion to define overflow water on the manuscript. Thus, I think it should be included in the manuscript and not in supplementary material. It also highlights the distribution of overflow water.
Response: Fig. S4 has been deleted from the Supplement and a new figure (Fig. 6) has been added to the revised main manuscript. The new figure is not identical to Fig. S4 since panel a of that figure would then be identical to the original Fig. 9a (new Fig. 11a). Also, this would require re-ordering some of the text. Instead, the information in the original Fig. S4 has been put into the form of a histogram, which ought to convey the original message in a better way.

Comment 2.3: Page 6 line 11. Maybe here it can be stated that even though throughout the manuscript the 3$^\circ$C isotherm is used as upper limit for the overflow water, at he end of section 3 a sensitivity analysis is carried out.
Response: Done. [page 6, line 13-15].

Comment 2.4: Figure 6 needs to be turn upside down so the upper side is Bottom temperature and the lower velocities to be consistent with the text and caption.
Response: Done for new Fig. 8

Comment 2.5: The caption of Figure 7 is also opposite to the figure and text.
Response: The caption of Fig. 7 (new Fig. 9) has been modified.

Comment 2.6: Figure 8. Could you also add a line for the 3$^\circ$C on the sections to compare with the 27.8 kg m-3 isopycnal. Could the map be slightly bigger so one can read the isobaths?
Response: Done in new Fig. 10. This also inspired a new comment on the difference between these two isopycnals [page 10, line 11-14].

Comment 2.7: Using the terminology of transport density when part of your data has density and part not, even if it has nothing to do, creates confusion. Better use the terminology of (volume) transport per unit length.
Response: Done throughout the manuscript.

Comment 2.8: Page 8 line 22 and Page 10, in lines 26 and 29, please add the uncertainty that these values have. Table 4 add uncertainty.
Response: Since our results showed a much weaker WV-overflow than expected, we have focused on estimating its maximum value, including uncertainties, rather than average values with uncertainties. We do, however, see the referee's point and have tried to do as requested where we find it possible. For the first case mentioned (original Page 8 line 22; new [page 8, line 26]), we have not been able to find any reasonable way to define or estimate uncertainty, but this value (1.5 m$^2$ s$^{-1}$) is not used further and does not affect the main conclusions of the manuscript. For the second case (original Page 10, lines 26 and 29), we are able to estimate an uncertainty value and

ought to have done that originally. This is now done on [page 9, line 27-29] and [page 11, line 10-14]. For that purpose, we added text to Sect. 2.4 [page 5, line 30 to page 6, line 2] and a reference [page 16, line 31-32]. For consistency, we also introduced an uncertainty for the width of the overflow layer [page 6, line 25]. The abstract has been updated accordingly [page 1, line 18] as has the conclusion [page 13, line 28]. For Table 4, we have also added uncertainties and modified the table caption accordingly.

Comment 2.9: I think that breaking the author breaks the flow of the reading by trying to have small paragraphs. I think the following paragraphs pairs can benefit from blending into one: (1) starting on page 2 from line 30, to page 3 in line 7, (2) page 3 starting on line14 and ending in line 23, (3) page 3 starting on line 24 and ending in line 32, (4) page 10 from line 17 to line 23
Response: Done. [page 2, line 32], [page 3, line 17], [page 3, line 28], [page 11, line 3].

Comment 2.10: Page 6, Line 4 remove: "which appears to be", is it or is it not?
Response: Done. [page 6, line 6].

Comment 2.11: Figure 5. Please advise the reader that each figure has a different y axis on the caption.
Response: Done in caption of new Fig. 7.

**Anonymous Referee #3**
Comment 3.1: Lines 14-19, p.6: I think that the different distributions at A and C suggest the two sites possibly have different mechanisms going on. A seems to be almost normally distributed around about 3$_{\circ}$C – sometimes it has colder water, sometimes warmer. I wondered if it was as a result of a change in the lateral or vertical position of the overflow interface depending on the volume of overflow (or other effects). In contrast C has a bi-modal structure – it is either overflow water, or water centred upon 4$_{\circ}$C. I wondered whether this site is mostly within the overflow, but that it is sometimes replaced by (northward flowing?) Atlantic water?
Response: We agree and have tried to make this clearer in the text. [page 6, line 20-22].

Comment 3.2: Lines 5-6 p.7: suggest addition that U(T) and T(A) are positively correlated, whilst U(T) and T(B) are negatively correlated.
Response: Text has been added, which clarifies that the two correlations have opposite signs. [page 7, line 10].

Comment 3.3: Lines 1-8, p.7: think you need to add a sentence to remind the readers the velocity measurements are at B only. The velocity could change quite a lot laterally? I think I find it most surprising that U(D) and T(B) are not significantly correlated!
Response: A sentence has been added on lateral velocity variation [page 7, line 12-14]. As to the correlation between U(D) and T(B), it is in fact significant in Table 2, although weakly. Perhaps, the referee intended to refer to the correlation between U(D) and T(A).

Comment 3.4: Lines 12-13 p.11: could you add an additional sentence or two explaining how you calculated the uncertainties associated with the regression coefficients?
Response: An explanation (referring to Sect. 2.4) has been added to the caption of the new Fig. 13 where the uncertainties are shown.

Comment 3.5: Lines 18-23, p.12: your figure 1b also suggests there may be other places along the IFR where overflow is stronger than through the WV.
Response: A reference to the original Fig. 1b (new Fig. 2b) has been added to the text. [page 13, line 5-6].

Comment 3.6: I found some of the figure captions difficult to follow. For some figures (e.g. 6) you had: (a) description, (b) description; whilst for others (e.g. 7) you have: description (a), description (b). I think sticking to one format would help the reader.
Response: The captions of the new Figs. 2, 9, 11, 13, and 15 have been modified so that they all have the same format.

Comment 3.7: Figure 4: could you define BTL in figure caption (for those of us who look at the figures before reading the article)
Response: Done in caption of new Fig. 5.

Comment 3.8: Figure 4: suggest addition of where the CTD stations were (e.g. by crosses on upper x-axis)

Response: Done in new Fig. 5.

Comment 3.9: Figure 5: either plot all on same y-axis, or put note to reader in caption that different y-axis are used.
Response: Done in caption of new Fig. 7.

Comment 3.10: Figure 6: swop round so (a) is top panel, and (b) bottom panel
Response: Done in new Fig. 8.

Comment 3.11: Figure 9: could you add the locations of A and C onto the plots too? I think this would be interesting to see
Response: Done in new Fig. 11.

Comment 3.12: Table 1: were all the instruments 17 m off bottom? (as mentioned in the text)
Response: Clarifying text has been added to the caption of Table 1.